# Knowledge, Attitude, and Practices of the General Population toward the Old-New Outbreak of Cholera in a Developing Country

**DOI:** 10.3390/tropicalmed8040236

**Published:** 2023-04-20

**Authors:** Marwan Akel, Fouad Sakr, Chadia Haddad, Aline Hajj, Hala Sacre, Rony M. Zeenny, Jihan Safwan, Pascale Salameh

**Affiliations:** 1School of Pharmacy, Lebanese International University, Beirut P.O. Box 146404, Lebanon; 2INSPECT-LB (Institut National de Santé Publique, d’Épidémiologie Clinique et de Toxicologie-Liban), Beirut P.O. Box 12109, Lebanon; 3School of Education, Lebanese International University, Beirut P.O. Box 146404, Lebanon; 4UMR U955 INSERM, Institut Mondor de Recherche Biomédicale, Université Paris-Est Créteil, 94010 Créteil, France; 5École Doctorale Sciences de la Vie et de la Santé, Université Paris-Est Créteil, 94010 Créteil, France; 6School of Medicine, Lebanese American University, Byblos P.O. Box 13-5053, Lebanon; 7Research Department, Psychiatric Hospital of the Cross, Jall Eddib P.O. Box 60096, Lebanon; 8Faculté de Pharmacie, Université Laval, Québec G1V 0A6, Canada; 9Oncology Division, CHU de Québec Université Laval Research Center, Québec G1S 4L8, Canada; 10Laboratoire de Pharmacologie, Pharmacie Clinique et Contrôle de Qualité des Médicament, Faculté de Pharmacie, Université Saint-Joseph de Beyrouth, Beirut P.O. Box 1107-2180, Lebanon; 11Drug Information Center, Order of Pharmacists of Lebanon, Beirut P.O. Box 11-2807, Lebanon; 12Department of Pharmacy, American University of Beirut Medical Center, Beirut P.O. Box 11-0236, Lebanon; 13Faculty of Pharmacy, Lebanese University, Hadath P.O. Box 6573-14, Lebanon; 14Department of Primary Care and Population Health, University of Nicosia Medical School, Nicosia 2417, Cyprus

**Keywords:** cholera, waterborne, foodborne, knowledge, attitude, practice, lower-middle-income country, refugee, infection

## Abstract

**Background:** In October 2022, the first case of cholera since 1993 was recorded in Lebanon. This study aimed to develop and validate a tool to explore the knowledge, attitudes, and practices (KAP) toward cholera infection and its prevention among the general population in Lebanon and identify the associated factors related to the KAP assessment to guide prevention and awareness strategies. The nation’s already precarious healthcare system might become overwhelmed by the response to the cholera outbreak. Therefore, evaluating the level of cholera-related KAP among the Lebanese population is crucial since it directly affects the disease’s treatment, control, and prevention. **Methods:** This online cross-sectional study was carried out between October and November 2022 during the cholera outbreak in Lebanon. Snowball sampling was used to recruit 448 adults living in Lebanon. **Results:** The suggested KAP scales had adequate structural and convergent validity and were internally consistent. The disease knowledge was inversely associated with the reluctance to receive educational information (β = −1.58) and cigarette smoking (β = −1.31) but positively associated with the female gender (β = 1.74) and awareness of vaccine availability and efficacy (β = 1.34). For attitude, healthcare professionals were less frightened than others (β = 2.69). Better practices were related to better knowledge (β = 0.43), while inadequate practices were associated with getting information from social media (β = −2.47). **Conclusions:** This study could identify notable gaps in the knowledge, attitudes, and practices, which varied according to participant characteristics. Cholera incidence can be reduced by improved community education and training, increased access to safe water, sanitation, and hygiene amenities, and changes in behavior. These findings warrant additional actions by public health stakeholders and governmental authorities to promote better practices and curb disease transmission.

## 1. Introduction

Cholera is an acute diarrheal illness brought on by an infection with the Vibrio cholerae bacterium [1,2]; it occurs after the ingestion of contaminated food or water. Without prompt and effective rehydration, cholera can cause severe dehydration and death. Rapid assays are utilized for a provisional diagnosis of cholera, with culture-based organism isolation serving as the confirmatory method [3]. People with the cholera bacteria can spread cholera outbreaks by shedding billions of the contagious *Vibrio cholerae* bacteria. The cornerstone of treatment consists of fluids and electrolyte administration, with antibiotics being given in cases of severe dehydration [1].

The recommended cholera disease prevention and control measures include the provision of safe drinking water and adequate sanitation to at-risk populations, as well as prompt and appropriate healthcare for individuals presenting with clinical disease. Additionally, oral cholera vaccines can play a critical role in the control of the disease [4,5,6].

Recent reports have demonstrated that cholera is a global public health issue and a sign of social injustice affecting between 1.4 and 4.3 million people annually and can result in up to 143,000 deaths worldwide [7]. Cholera outbreaks affected multiple countries over the past years, mainly in Asia and Africa, whereby 323,369 cases and 857 deaths were notified from 24 countries in 2020 [8].

On 6 October 2022, Lebanon recorded its first case of cholera since 1993, likely resulting from the outbreak in Syria crossing the porous border between the countries. As of 10 November 2022, 3160 confirmed/suspected cases and a total of 18 related deaths have been recorded across the nine governorates of Lebanon (Beirut, Akkar, North, Keserwan-Jbeil, Mount Lebanon, Beqaa, Baalbek-Hermel, South, and Nabatieh) [9]. A three-year financial crisis and the massive explosion of the port of Beirut in August 2020, which destroyed essential medical infrastructure in the capital city, have both severely harmed the healthcare system in Lebanon. According to the United Nations High Commissioner for Refugees (UNHCR), Lebanon is home to the highest number of refugees in the world per capita and square kilometer, with 1.5 million Syrian refugees and approximately 13,715 refugees of other nationalities. Moreover, the already precarious healthcare system has been strained by power outages, water restrictions, and severe inflation. Subsequently, many Lebanese families now routinely limit water due to poverty, as they cannot afford to own individual water tanks for consumption and domestic usage.

Immediate action is thus necessary to halt the cholera outbreak from putting the lives of children and families at risk since the nation’s water management and healthcare systems have been detrimentally impacted by the escalating economic crisis and persistent statewide power shortages [10]. As part of a comprehensive response to cholera prevention and control, the Lebanese Ministry of Public Health has launched a cholera vaccination campaign as an immediate short-term intervention to help control the spread of cholera. This campaign targeted all refugees and host communities aged one year and above [11].

Another factor that contributes to the spread of cholera is the poor knowledge and awareness of the public about cholera’s modes of transmission and early methods of detection and treatment of its symptoms [12]. Studies on knowledge, attitude, and practices have been conducted on many infectious diseases, such as bacterial infections [13] and COVID-19 [14].

A study conducted in Lebanon [15] showed some correlates related to cholera knowledge, attitude, and practices (KAP); its findings could not provide a clear picture of the overall correctness of the KAP due to the absence of standardized results. Furthermore, the survey tool that was used was not validated, which could increase the risk of information bias, and the conclusion regarding practice could be subject to confounding in the absence of multivariable analysis. Thus, more specific and better-defined measures should be used to locate the weaknesses in the cholera-related KAP among people residing in Lebanon.

Therefore, this study aimed to develop and validate a tool to explore the KAP toward cholera infection and its prevention among the general population in Lebanon and identify the associated factors related to the KAP assessment to guide prevention and awareness strategies.

## 2. Materials and Methods

### 2.1. Study Design

This cross-sectional study was carried out between October and November 2022, during the cholera outbreak in Lebanon, using an online questionnaire to reach a broader population, including people in remote areas where cholera was first discovered, and achieve greater statistical power and sample representativeness. The survey tool was shared with a first sample selected from different regions of the nine governorates of Lebanon (Beirut, Akkar, North, Keserwan-Jbeil, Mount Lebanon, Beqaa, Baalbek-Hermel, South, and Nabatieh). This snowball sampling technique was used to recruit 448 participants. Respondents were briefed about the topic and the different aspects of the questionnaire and were assured of the anonymity of their responses at the beginning of the questionnaire before filling it out. All adults aged 18 years and more living in Lebanon were eligible to participate.

### 2.2. Ethical Aspect

The Lebanese International University School of Pharmacy Research and Ethics committee approved the study protocol (2022RC-041-LIUSOP). Written consent was obtained from each person at the beginning of the questionnaire.

### 2.3. Sample Size Calculation

The CDC Epi-info software was used to calculate the minimum sample size. The expected frequency was kept at 50% to yield the largest sample size. Accordingly, a minimum of 384 participants was required to produce a 95% confidence interval, with a 5% alpha error and a power of 80%.

### 2.4. Questionnaire and Variables

The questionnaire used was in Arabic (the native language in Lebanon) and English, both languages being widely spoken in the country. It included a sociodemographic category and a scale-based section related to knowledge, attitude, and practices toward cholera.

#### 2.4.1. Sociodemographic Data

In this part of the questionnaire, participants were asked about their general sociodemographic data, including age, gender, nationality, area of residence, educational level, income, marital status, smoking and alcohol consumption, health status and diseases, being or having a health professional member in the family, healthcare access, and information about house crowding index (including number of children/dependents).

#### 2.4.2. Scale-Based Variables

The following scales were used in the questionnaires:
*The InCharge Financial Distress-Financial Well-Being (IFDFW) Scale:*


This eight-item tool is a validated subjective measure of financial distress and financial well-being. It depicts a spectrum ranging from negative to positive feelings and responses toward one’s financial situation that might affect KAP toward cholera. This scale is of particular importance in the current context of the severe Lebanese socioeconomic crisis. Responses were rated on a scale from 1 to 10, with higher scores indicating better financial well-being [16]. Cronbach’s alpha of the scale among our sample was 0.914.


*The Knowledge, Attitude, and Practice scales:*


In the absence of previously validated scales regarding the studied concepts, questions regarding knowledge, attitude, and practices were developed based on previous studies [17,18,19,20,21,22,23]; additional questions were added when deemed necessary. The KAP scales were adopted since it has been suggested that cholera prevention, control, and management directly depend on individual and community sanitation knowledge, behaviors, attitudes, and practices. Using a Delphi consensus, the series of questions was content validated by six of the authors, who are public health experts. Of those, five are pharmacists, among whom one is an infectious diseases specialist. An agreement of more than 90% of the authors was necessary to keep an item. The list of items was then ready to be administered to the study participants.

Knowledge about cholera:

This section included 5 subsections, i.e., symptoms of cholera, causes/etiologies of cholera, prevention knowledge, sources of information, and vaccine knowledge. Answer options to all the questions were yes/no/I don’t know, except for the last question regarding the dosage form of the cholera vaccine. A higher score indicated better knowledge. Cronbach’s alpha in this study was 0.820.

Attitudes toward cholera:

This section included 24 items that reflect the attitude of the public toward cholera. Each statement had 5 answer options ranging from strongly disagree to strongly agree. A higher score indicated a less frightful and more careless attitude. Cronbach’s alpha was 0.815.

Practices about cholera:

The last section of the questionnaire included 19 questions about the practices toward cholera. Each statement had 5 answer options ranging from strongly disagree to strongly agree. A higher score indicated a better application of preventive measures. Cronbach’s alpha in this study was 0.794.

#### 2.4.3. Translation Procedure

One of the authors, who is a senior research member, a pharmacist, and a medical editor, evaluated and verified the translation (from English to Arabic) that was performed by one of the collaborators while developing the questionnaire. Discrepancies were solved by consensus among all the authors, who are all Arabic and English speakers. The final questionnaire was piloted on twenty bilingual participants to assess the clarity of questions.

### 2.5. Statistical Analysis

Data were analyzed on SPSS software version 25. A descriptive analysis was performed using absolute frequencies and percentages for categorical variables and means and standard deviations (SD) for quantitative measures.

Construct validity of the knowledge, attitude, and practice scales was assessed using factor analysis with principal component analysis (PCA). The Promax rotation method was used, given that factors were correlated. The Kaiser–Meyer–Olkin (KMO) and Bartlett’s test of sphericity was calculated to ensure the model’s adequacy. Factors with Eigenvalues greater than one were retained, and the scree plot method was used to determine the number of components to extract [24]. Furthermore, factors related to KAP were also correlated to assess convergent validity. The internal consistency of the studied scales was assessed using Cronbach’s alpha: values of α ≥ 0.7 and ≥0.8 were considered acceptable and excellent, respectively [25]. Before generating the scores, items’ codes with inverse directions were reversed to allow for correct scoring and a smoother interpretation of knowledge, attitude, and practice concepts.

The KAP variables were deemed normally distributed, as verified by visual inspection of the histogram, while the skewness and kurtosis were below |1.96|. In the bivariate analysis, the independent-sample T-test was used to compare the means of the KAP scales between two groups, whereas the ANOVA test was used to compare three or more means. The Pearson correlation test was used to correlate continuous variables. A *p*-value less than 0.05 was considered significant.

After that, three linear regressions using the stepwise method were performed, taking the KAP scales as the dependent variables. In the first linear regression, the knowledge scale was taken as the dependent variable and sociodemographic characteristics as the independent variables. In the second linear regression, the attitude scale was taken as the dependent variable, and knowledge and sociodemographic characteristics as the independent variables. In the third linear regression, the practice scale was selected as the dependent variable, and knowledge, attitude scales, and sociodemographic characteristics as the independent variables. Variables that showed a *p*-value < 0.2 in the bivariate analysis were included in the multivariable models to decrease potential confounders. When applicable, the normality of the scales used as dependent variables was verified by the normality line of the regression plot and the scatter plot of the residuals.

## 3. Results

### 3.1. Description of Sociodemographic Characteristics

Table 1 displays sociodemographic and other participant characteristics. Most participants were females (73.4%), single (68.5%), healthcare professionals (70.5%), employed (51.8%), had a university education level (97.3%), an intermediate or high income (57.6%), and lived in an urban area (72.0%), particularly Beirut and Mount Lebanon (74.3%). Regarding the heads of the family, 70.8% had a university education level and 81.3% were employed. Almost half of the participants belonged to the upper-middle social class (45.8%) and were full-time employed (40.2%). Only 14.1% were current cigarette smokers, 20.3% were narguileh smokers, and 26.8% drank alcohol. Moreover, 17.6% had a chronic disease, and 83.5% had easy access to healthcare. The mean age of the participants was 29.52 ± 11.03 years (Min 18; Max 96), the mean household crowding index was 0.98 ± 0.52, and the mean financial well-being was 43.06 ± 17.13.

### 3.2. Factor Analysis of the KAP toward Cholera

The knowledge scale items produced three factors with an eigenvalue over 1, accounting for a variance of 42.08% (Bartlett sphericity test *p* < 0.001; KMO = 0.861; Cronbach’s alpha = 0.820). Factors 1, 2, and 3 were about false information (to be disproved by participants), knowledge of preventive measures, and cholera-related symptoms, respectively. Regarding the attitude, five factors were yielded with a total variance of 71.91% (Bartlett test of sphericity *p* < 0.001; KMO = 0.922; Cronbach’s alpha = 0.815). Factors were was about fear feelings (Factor 1), psychosomatic symptoms (Factor 2), trust in public health containment of cholera (Factor 3), distrust in the health system and fear of dying (Factor 4), and false beliefs and myths (Factor 5). The practice scale produced three factors accounting for a variance of 64.77% (Bartlett test of sphericity *p* < 0.001; KMO = 0.925; Cronbach’s alpha = 0.794) (Table 2); its related factors included correct preventives measures (Factor 1), risky behaviors (Factor 2), and inadequate practices (Factor 3).

### 3.3. Description of the KAP Scales

Table 3 describes the median, mean, SD, and range of the KAP scales used in this study. According to the percentage mean, the majority of the participants had good knowledge (70.54%) and correct practice (70.91%) toward cholera, but 71.43% had a careless attitude.

### 3.4. Structural and Convergent Validity Analysis

Table 4 presents structural and convergent validity analyses. Knowledge of preventive measures did not significantly correlate with false information, while knowledge about prevention positively and moderately correlated with symptoms-related knowledge. A weak but positive correlation between false information and symptoms-related knowledge was also noted. For attitude, all fear-related and false beliefs factors positively and significantly correlated with each other, except for the trust in public health containment measures that negatively correlated with all other factors. As for practice, risky behaviors were inversely related to correct preventive measures, while wrong behaviors were weakly but positively associated with correct preventive measures and better correlated with risky behaviors. Moreover, while attitude and knowledge are not significantly related, better knowledge was significantly associated with better practice, while a more careless attitude was significantly associated with worse practice (Table 4).

### 3.5. Bivariate Analysis

Higher knowledge scores were significantly associated with being a healthcare professional, female gender, being married, the head of the family having a university education level, being employed, having a high income, working full time, having easy access to healthcare, knowing that the cholera vaccine is available, and not smoking cigarettes. Moreover, a higher household crowding index was significantly associated with a lower knowledge score.

A more careless attitude score was significantly associated with being a healthcare professional. Taking the practice scale as the dependent variable in the bivariate analyses showed that being a female and knowing that the cholera vaccine is available were significantly associated with higher practice scores. Furthermore, a higher knowledge score was significantly associated with better practice, while a more careless attitude was significantly associated with poorer practice (Table 5).

No association was found between age and KAP (*p* > 0.05 for all). The age variable was divided into quartiles (Category 1 below 22 years, Category 2 between 23–25 years, Category 3 between 26–35 years, and Category 4 above 36 years), and bivariate analysis was conducted between age categories and KAP. The results showed a significant association between education level and attitudes and practices in participants below 22 years. A significantly higher mean attitude score was found in those with a school education level as compared to a university education level (80.50 vs. 72.09, *p* < 0.001), while a higher mean practice score was found in those with a university education level as compared to school education level (63.27 vs. 54.50, *p* < 0.001). The association between knowledge scale and education level tended to be significant in participants over 36 years (*p* = 0.055). No significant association was found between education level and KAP in other age categories.

The association between the source of information about cholera and the KAP scales is presented in Table 6. The results showed that higher knowledge scores were significantly associated with obtaining information from electronic databases, from the pharmacist, from the Ministry of Public Health (MOPH) brochures, and conferences. Low knowledge score was significantly associated with those refusing to seek such information.

More careful attitude scores were significantly associated with obtaining information from social media platforms and the MOPH brochures. However, a more careless attitude score was significantly associated with those refusing to seek such information.

Higher mean practice scores were significantly associated with those obtaining the information from the Ministry of Public Health brochures. However, lower mean practice scores were significantly associated with those refusing to seek such information and those obtaining the information from newspapers.

### 3.6. Multivariable Analysis

A first linear regression taking the knowledge scale as the dependent variable showed that the female gender (β = 1.74), easy access to healthcare (β = 2.05), working as a full-time (β = 1.24), being a healthcare professional (β = 2.50), being a health student (β = 2.39), knowing that cholera vaccine is available and effective (β = 1.64), and obtain information about cholera from electronic databases (β = 1.19) were significantly associated with a higher knowledge score. However, refusing to seek information (β = −1.58) and smoking cigarettes (β = −1.31) were significantly associated with lower knowledge scores (Table 7, model 1).

A second linear regression taking the attitude scale as the dependent variable showed that being a healthcare professional (β = 2.69) was significantly associated with a careless attitude. However, getting information about cholera from social media (β = −3.29) and MOPH brochures (β = −3.60) and having a higher household crowding index (β = −2.24) were significantly associated with a more careful attitude (Table 7, model 2).

A third linear regression taking the practice scale as the dependent variable showed that better knowledge (β = 0.43), being a female (β = 2.55), knowing that the cholera vaccine is effective (β = 3.17), and obtaining information from MOPH brochures (β = 2.41) were significantly associated with a better practice score. However, a higher attitude score (β = −0.27) and obtaining information from the newspaper (β = −3.37) or social media (β = −2.47) were significantly associated with lower practice scores (Table 7, model 3).

## 4. Discussion

This study evaluated KAP toward an emerging outbreak of Vibrio cholera in Lebanon, where the situation in Lebanon should not be different from that of other developing countries, as the water management and healthcare systems have been severely impacted by the deepening economic crisis, continuous nationwide power shortages, and the high numbers of refugees and displaced populations living in precarious conditions [26]. The results showed that gender, knowing about vaccine availability and efficacy, access to healthcare, occupation, and source of information were predictive of better cholera knowledge. However, smoking and reluctance to seek information about cholera were associated with lower knowledge. A more careless attitude toward cholera was predicted by occupation, whereas a less careless attitude was predicted by the source of information and a higher household crowding index. Better knowledge of cholera was significantly associated with better practices, and a more careless attitude was associated with worse practices. Furthermore, worse practices were predicted by the source of information about the infection.

Good overall knowledge about cholera (70% on average) was found, indicating better cholera knowledge in the Lebanese community in comparison to other countries that had to deal with this outbreak [22,27]. A better knowledge score was significantly predicted by better knowledge of the cholera vaccine availability and efficacy and easy access to healthcare. This finding reveals the imperative role of the Lebanese healthcare system in raising awareness and providing proper education to the community about the preventive and treatment measures for cholera. Our results are consistent with previous findings that reported better knowledge of cholera with appropriate exposure to a medical and healthcare environment [28].

One study reported poor overall knowledge of cholera among non-trained Nigerian healthcare providers [29]. Although the current outbreak is novel and non-traditional in Lebanon, our findings show that the healthcare system and healthcare education are well competent to deal with this unprecedented situation. This fact was determined by a significantly higher score of knowledge among healthcare workers and students. The sociodemographic characteristics of the community also appear to be influential predictors of cholera knowledge. Indeed, females had better knowledge, while smokers had worse knowledge. It is not fully understood why females had a better understanding of the outbreak compared to males. One hypothesis is that females are more concerned about their health and that of their family, and thus, they are more inclined to acquire ampler information about the disease. Our results are consistent with other findings from the Kingdom of Saudi Arabia, showing a significantly improved knowledge and practice toward cholera infection among females [18]. Furthermore, smokers were previously reported to have different personality traits and neuroticism compared to non-smokers, which could affect their knowledge and behavior about their health conditions, including cholera [30].

Obtaining information about cholera and the source of information appear to be important predictors of knowledge about the infection. Significantly lower knowledge scores were found among individuals who refused to seek information about cholera. Generally, less information seeking is reportedly associated with lower health literacy on various communicable and non-communicable illnesses [31,32,33]. Furthermore, a better knowledge score was significantly associated with getting information on cholera from electronic databases. The role of electronic databases in promoting the health information system is well established. Digital data sources, including websites, were shown to be essential for collecting and mining information in the area of infectious diseases and early warning systems [34].

A recent study explored the attitude of the Lebanese community toward cholera [15]. Its main drawback is that it assessed knowledge of the transmission mode and causes of the infection rather than attitudes toward the disease and did not use validated scales. Hence, comparing our results is difficult, except for the association of knowledge with being a healthcare professional. Regarding attitude and practice, the difference in the scope of questions between our study and theirs hampers any comparison. The current study provides better evidence of the factual attitude of Lebanese residents toward cholera. It revealed mixed overall attitudes of the community about the infection, with a mixture of careless and frightful attitudes unrelated to knowledge but inversely associated with preventive measures. Previous studies from Bangladesh and Tanzania also reported overall positive attitudes toward cholera [20,22]. However, none of these studies transformed the attitude items into a validated scale to determine the predictors of attitude.

Our results showed that healthcare professionals had a significantly more careless attitude, likely due to the over-reassuring nature of their education; moreover, attitudes might not necessarily correlate with practices among healthcare professionals [35]. However, in Kenya, a more positive attitude toward cholera was significantly associated with receiving some formal education [21]. More studies are necessary to explain this finding.

An inverse association was found between careless attitude and obtaining information from both official (MOPH) and non-official (social media) sources. Therefore, nationwide campaigns should raise awareness to educate and reassure the community about the disease while keeping a reasonable margin of fear and concern to improve attitudes toward preventive measures. It is also vital to raise awareness about the reliable sources of information on the internet to mitigate misinformation, which is frequent on social media. In other countries, combating misinformation was pivotal to the success of cholera information campaigns [36,37,38].

To the best of our knowledge, no previous studies from other countries explored the impact of the house crowding index on attitudes toward cholera, likely because most of the literature on cholera has been carried out in underdeveloped countries. In the present study, a higher house crowding index was negatively associated with a careless attitude toward cholera but was not associated with better preventive practices. Lebanon is a developing lower-middle-income country that has been facing a dramatic socioeconomic crisis over the past three years. This overwhelming crisis has been linked to higher rates of poverty and catastrophic sociodemographic and medical consequences [39]. A higher house crowding index reflected a lower socioeconomic status that was correlated to more fearful attitudes toward cholera but was not associated with better preventive measures. A possible explanation could be poor health education and improper water and sanitation resources among this category of the population [40], who cannot take the appropriate preventive measures despite their careful attitude.

Our findings showed good overall practices related to cholera. Previous studies from other countries have described single-entity items for practices toward cholera. In Tanzania, water treatment with chlorine was rarely applied to prevent infection [41]. The use of oral rehydration solution (ORS) in the management of acute illness to prevent dehydration among their population was also scarce. Another study from Kenya reported practice gaps related to toileting and personal hygiene [21]. Furthermore, better knowledge among the Kenyan population was not translated into better practices. The current study adds to the literature a comprehensive tool to assess practices on preventive and treatment measures for cholera. This tool also helped determine the independent predictors of practice. Better practices were significantly associated with better knowledge and a more careful attitude, consistent with other findings from Bangladesh showing an association between better practices and a higher level of knowledge about cholera [42]. Additionally, a worse practice score was significantly associated with obtaining information from social media. Indeed, social media could be a source of inappropriate health education. A previous report from Iraq determined that social media is a major source of wrong knowledge and practices for the vast majority of the population on cholera [23].

### 4.1. Practical Implications

Cholera continues to be a global public health concern, with over 1.3 billion people at risk of infection and 2.9 million cases occurring annually in countries with the endemic [43]. Since the outbreak was declared in Lebanon, the United Nations International Children’s Emergency Fund (UNICEF) has increased its activity to help the Lebanese government by securing chlorine for water disinfection, fuel to operate wastewater treatment and lifting stations, and conducting awareness campaigns to combat misinformation [26].

Rapid-spreading cholera outbreaks are typically transmitted through contaminated water supplies. Increased access to safe water, sanitation, and hygiene amenities, as well as behavioral changes brought about by community education and training, can help to minimize cholera incidence [44]. Therefore, the assessment of the community KAP is crucial to limit the incidence of infection and control the outbreak. This study created three tools to assess knowledge, attitude, and practice toward cholera. The content and structure validity of each tool was established, and their reliability was determined by their high Cronbach’s alpha. These novel tools could help practitioners, researchers, and other stakeholders assess the overall KAP toward cholera and identify areas for action based on independent factors that can correlate with poor KAP. These findings provide information for public health stakeholders to enforce community education to achieve better KAP and minimize disease transmission.

Our findings identified notable gaps in the Lebanese community linked to poor knowledge due to the reluctance to seek information about the outbreak, negative attitudes despite obtaining information from official sources and social media, and worse practices despite positive attitudes. Consequently, targeted health education campaigns, public awareness programs, and strategies to combat misinformation and disinformation on social media are warranted. It is also essential to increase access to digital resources and extend correct information to push-type social media to increase public awareness and trigger behavioral changes. Comprehensive national strategies related to health literacy promotion are thus necessary.

However, in areas with a higher house crowding index, improving water and sewage infrastructures is expected to be more effective in controlling enteric infections than enhancing KAP. Furthermore, facilitating access to clean water at affordable prices is a national emergency that complements the abovementioned actions.

### 4.2. Strengths and Limitations

The current study has several strengths. The data were collected from districts all over Lebanon, which allowed for assessing the KAP toward cholera of the entire Lebanese community regardless of the sporadic outbreak regions. The sample size was adequate for all statistical analyses to be carried out and to determine the variables associated with better or poor KAP. To our knowledge, these are the first tools related to cholera KAP to be ever validated in a current complex crisis setting, combining socioeconomic hardship and sanitary decay. This fact might increase the usefulness of the suggested tools in developing countries with similar circumstances. The content and construct validity of the KAP tools were also confirmed, and the tools appeared consistent and reliable, which is expected to consolidate the validity of the results and decrease information bias.

Nevertheless, several limitations could not be avoided in this study. The cross-sectional design does not provide temporality but remains practical for capturing a snapshot amid outbreaks and rapidly identifying gaps, although causal associations cannot be confirmed. Furthermore, data were collected using a snowball technique (online questionnaire), which is a nonrandom technique that could have been associated with a possible selection bias as this data collection technique might have eliminated people with poor digital literacy. Nevertheless, it is believed that the likelihood of this bias is minimal as the survey was both in Arabic, the natively spoken language of Lebanon, and English, and almost all the people residing in Lebanon hold smartphones and are familiar with online access. However, some other people who were not interested or cooperative in these kinds of surveys were not enrolled in the study, which might have affected the results, as a possible correlation might be found between those interested and concerned about infectious diseases and KAPs and those who were not. Thus, the results obtained can underestimate the correct cholera-related knowledge, attitude, and practices of the population.

Moreover, the results of the current study are not representative of the entire Lebanese population, as the sample had an unequal gender distribution, with the majority being females. In addition, the study did not explore the impact of mental health on KAP toward cholera as this aspect could provide a more holistic understanding of the factors influencing individuals’ KAP and help inform targeted interventions for different sub-groups within the community. Further research is recommended to explore the effect of mental health parameters on cholera KAP and outcomes in the context of severe sanitary, economic, and social crises in Lebanon.

Finally, residual confounding related to mental health cannot be precluded, as the present study did not determine the impact of mental health on cholera KAP. Moreover, variables related to the environment and urban and rural settings would better be included to further minimize residual confounding.

## 5. Conclusions

This study could develop and validate reliable tools to measure cholera-related knowledge, attitude, and practice; these tools would be helpful in other developing countries of similarly challenging socioeconomic and sanitary contexts. This study also identified notable gaps in the disease knowledge associated with the reluctance to receive educational information and cigarette smoking. Participants who obtained information from official sources and social media and those with low socioeconomic status (reflected by a higher house crowding index) had fearful attitudes; however, these attitudes did not translate into better preventive practices. Better practices were related to better knowledge, while inadequate practices were associated with a less careful attitude and getting information from social media. These findings warrant additional actions by public health stakeholders and governmental authorities to promote better practices and curb disease transmission. Comprehensive and multifaceted strategies, developed through incorporating approaches targeting knowledge, attitudes, and practices and considering socioeconomic factors and information sources, are recommended to promote overall health literacy related to cholera and other infectious diseases.

## Figures and Tables

**Table 1 tropicalmed-08-00236-t001:** Sociodemographic and other characteristics of the studied population (N = 448).

	Frequency	Percentage
**Gender**		
Female	329	73.4
Male	119	26.6
**Marital status**		
Single/Widowed/divorced	307	68.5
Married	141	31.5
**Current governorate**		
Beirut	148	33.0
Mont Lebanon	185	41.3
North	47	10.5
South	50	11.2
Beqaa	18	4.0
**Education level**		
School level	12	2.7
University level	436	97.3
**Education level of the head of the family**		
School level	131	29.2
University level	317	70.8
**Occupation**		
Unemployed	216	48.2
Employed	232	51.8
**Occupation of the head of the family**		
Unemployed	84	18.8
Employed	364	81.3
**Household monthly income**		
No income	120	26.8
Low income: below 6 million Lebanese Lira (LBP)	70	15.6
Intermediate income: 6–18 million LBP	115	25.7
High income: over 18 million LBP	143	31.9
**Living place**		
Urban	323	72.1
Rural	125	27.9
**Housing condition**		
Very clean/Relatively clean	397	88.6
Very dirty/Relatively dirty	51	11.4
**Being a healthcare professional**		
Yes	316	70.5
No	132	29.5
**Having a healthcare professional in the family**		
Yes	222	49.6
No	226	50.4
**Work status**		
Does not apply	185	41.3
Full-time	180	40.2
Part-time	83	18.5
**Social class**		
Poor	10	2.2
Lower middle class	151	33.7
Upper middle class	205	45.8
Wealthy	8	1.8
I prefer not to answer	74	16.5
**Cigarette Smoking status**		
No	374	83.5
Yes, currently a smoker	63	14.1
Yes, previous smoker	11	2.5
**Nargileh smoker**		
No	342	76.3
Yes, currently a smoker	91	20.3
Yes, previous smoker	15	3.3
**Alcohol**		
No	312	69.6
Yes, currently	120	26.8
Yes, previous alcohol consumer	16	3.6
**Having a chronic disease**		
Yes	79	17.6
No	369	82.4
**Easy access to healthcare**		
Yes	374	83.5
No	74	16.5
	**Mean**	**SD**
**Age**	29.52	11.03
**Body mass index (BMI)**	24.37	8.19
**Household crowding index**	0.98	0.52
**Financial well-being scale**	43.06	17.13

**Table 2 tropicalmed-08-00236-t002:** Factor analysis of KAP scales.

**Factor Analysis of the Knowledge of Cholera**					
**Promax Rotated Matrix**					
**Factor**	**Factor 1** False Information		**Factor 2** Knowledge of Preventive Measures		**Factor 3** Cholera-Related Symptoms
Knowledge about prevention: respecting a safe distance from others (1.5 m)	0.687				
Knowledge about causes of respiratory droplets of infected persons	0.660				
Knowledge about sore throat symptoms	0.653				
Knowledge about prevention by applying insect repellent sprays	0.651				
Knowledge about prevention by avoiding close contact with people	0.649				
Knowledge about the causes of flies and mosquitoes	0.630				
Knowledge about ear pain symptoms	0.623				
Knowledge about the causes of Infected birds and chicken	0.613				
Knowledge about Runny nose symptoms	0.603				
Knowledge about prevention; Cholera cannot be prevented	0.526				
Knowledge about Bloody diarrhea symptoms	0.482				
Knowledge about causes: Other means	0.446				
Knowledge about Joint pain symptoms	0.439				
Knowledge about prevention: Washing hands thoroughly			0.641		
Knowledge about prevention: cleaning utensils/plates			0.624		
Knowledge about causes: Poor sanitation			0.623		
Knowledge about prevention: Washing vegetables and fruits			0.621		
Knowledge about the prevention of cooking food well			0.599		
Knowledge about causes: Poor hygiene			0.584		
Knowledge about prevention Applying hands sanitizers			0.563		
Knowledge about prevention: Covering food			0.498		
Knowledge about causes of Contaminated water			0.478		
Knowledge about causes Unwashed fruit/vegetables			0.478		
Knowledge about prevention Boiling or treating water			0.470		
Knowledge about prevention Using a latrine/avoiding open defecation			0.420		
Knowledge about causes of Contaminated food			0.408		
Knowledge about Watery diarrhea symptoms					0.719
Knowledge about Vomiting symptoms					0.702
Knowledge about symptoms of Nausea and decreased appetite					0.625
Knowledge about Abdominal pain symptoms					0.622
Knowledge about Dehydration symptoms					0.618
Knowledge about Malaise/lethargy symptoms					0.618
Knowledge about symptoms of fever					−0.555
**Percentage variance explained = 42.08**	19.61		14.89		7.57
**Cronbach alpha = 0.820**	0.849		0.734		0.712
**Kaiser–Meyer–Olkin** **(KMO) = 0.861**					
**Bartlett’s test of sphericity < 0.001**					
**Factor analysis of the Attitude toward Cholera**					
**Promax rotated matrix**					
**Factor**	**Factor 1** Fear feelings	**Factor 2** Psychosomatic symptoms	**Factor 3** Trust in Public Health containment of cholera	**Factor 4** Distrust in the health system and fear of dying	**Factor 5** False beliefs and myths
It makes me uncomfortable to think about cholera	0.978				
I am afraid of cholera	0.939				
I’m afraid of having a bad illness or being hospitalized because of cholera	0.851				
I become nervous or anxious when watching news and stories about cholera	0.851				
I am afraid of contagious diseases	0.788				
I’m afraid of losing my life because of cholera	0.743				
I’m afraid that anyone from my family or friends would get infected with cholera or get severely ill or die because of cholera	0.645				
I cannot sleep because I am worried about getting infected with cholera		0.941			
My heart races or palpitates when I think about cholera		0.936			
My hands get clammy when I think of cholera		0.794			
I don’t feel comfortable eating out because of cholera		0.358			
I think the government can contain the cholera outbreak			0.952		
I think repairing infrastructure can help contain the cholera outbreak			0.779		
I believe cholera vaccination should be encouraged/promoted			0.602		
I am worried that doctors and hospitals do not have sufficient medical supplies to handle the cholera outbreak				0.745	
I am worried that doctors and hospitals do not have sufficient knowledge and expertise to handle the cholera outbreak				0.708	
I am worried that laboratories and hospitals do not have the materials to diagnose cholera properly				0.705	
I would not travel to an area contaminated with cholera				0.644	
I am afraid of dying from cholera				0.422	
I think the cholera vaccine could cause harm					0.784
I think cholera is a myth					−0.734
I think hygiene measures are not effective in preventing cholera					0.718
I think I should live my life normally and it is God’s will if I ever catch the virus					−0.495
**Percentage variance explained = 71.91%**	41.66	14.24	6.66	4.87	4.45
**Cronbach alpha = 0.815**	0.937	0.814	0.807	0.817	0.577
**Kaiser–Meyer–Olkin (KMO)**	0.922				
**Bartlett’s test of sphericity**	<0.001				
**Factor analysis of the Practice regarding cholera**					
**Promax rotated matrix**					
**Factor**			**Factor 1** Correct preventive measures	**Factor 2** Risky behaviors	**Factor 3** Wrong practices
I wash fruits and vegetables with sanitized water			0.898		
I wash my hands thoroughly with soap			0.885		
I store water in clean and airtight bottles			0.885		
I collect the trash and dispose of it in the right place			0.845		
I drink bottled water			0.827		
I clean the toilets at home every day			0.820		
I cook food for a minimum of 15 min			0.783		
I go to the clinic/hospital if I suspect I have cholera			0.753		
I use oral rehydration solutions if I suspect I have cholera			0.737		
I boil or sanitize water with chlorine/house bleach			0.658		
I would take the cholera vaccine if it is available			0.631		
I order food from restaurants despite the cholera outbreak				0.811	
I eat in public places and restaurants despite the cholera outbreak				0.798	
I eat raw foods, such as fish, fruits, and vegetables				0.640	
I drink tap water					0.865
I wash fruits and vegetables with any available water					0.594
I use toilets in public places despite the cholera outbreak					0.531
I seek traditional medicines if I suspect I have cholera					0.500
**Percentage variance explained = 64.77%**			43.21	14.49	7.06
**Cronbach alpha = 0.794**			0.942	0.768	0.622
**Kaiser-Meyer-Olkin (KMO)**			0.925		
**Bartlett’s test of sphericity**			<0.001		

**Table 3 tropicalmed-08-00236-t003:** Description of the KAP toward cholera scales.

	Mean ± SD	Mean %	Median	Minimum	Maximum
Knowledge scale	23.28 ± 5.10	70.54%	24.00	0	31.00
Attitude scale	73.58 ± 12.30	71.43%	74.00	45.00	103.00
Practice scale	63.82 ± 10.97	70.91%	65.00	35.00	90.00

**Table 4 tropicalmed-08-00236-t004:** Cholera KAP scales structural and convergent.

**Cholera Knowledge Scale Structural Validity: Factor 1 included false information (to be disproved by participants), Factor 2 included preventive measures knowledge, and Factor 3 was about cholera-related symptoms**						
	**Knowledge Total Scale**	**Knowledge** **Factor 1**		**Knowledge Factor 2**		**Knowledge Factor 3**
Knowledge Factor 1	0.785			0.048		0.122
*p*-value	<0.001			0.309		0.010
Knowledge Factor 2	0.568	0.048				0.461
*p*-value	<0.001	0.309				<0.001
Knowledge Factor 3	0.602	0.122		0.461		
*p*-value	<0.001	0.010		<0.001		
**Cholera Attitude Scale Structural Validity: Factor 1 was about fear feelings, Factor 2 included psychosomatic symptoms, Factor 3 assessed trust in public health containment of cholera, Factor 4 is about distrust in the health system and fear of dying, and factor 5 included false beliefs and myths**						
	**Attitude Total Scale**	**Attitude Factor 1**	**Attitude Factor 2**	**Attitude Factor 3**	**Attitude Factor 4**	**Attitude Factor 5**
**Attitude Factor 1**	0.860		0.424	−0.622	0.719	0.095
*p*-value	<0.001		<0.001	<0.001	<0.001	0.044
**Attitude Factor 2**	0.675	0.424		−0.230	0.488	0.117
*p*-value	<0.001	<0.001		<0.001	<0.001	0.013
**Attitude Factor 3**	−0.351	−0.622	−0.230		−0.511	0.103
*p*-value	<0.001	<0.001	<0.001		<0.001	0.030
**Attitude Factor 4**	0.850	0.719	0.488	−0.511		0.155
*p*-value	<0.001	<0.001	<0.001	<0.001		0.001
**Attitude Factor 5**	0.324	0.095	0.117	−0.103	0.155	
*p*-value	<0.001	0.044	0.013	0.030	0.001	
**Cholera Practice Scale Structural Validity: Factor 1 included correct preventives measures, factor 2 included risky behaviors and factor 3 included wrong practices**						
	**Practice Total Scale**	**Practice Factor 1**		**Practice Factor 2**		**Practice Factor 3**
**Practice** Factor 1	0.874			−0.397		0.133
*p*-value	<0.001			<0.001		0.005
**Practice** Factor 2	0.020	−0.397				0.279
*p*-value	0.679	<0.001				<0.001
**Practice** Factor 3	0.422	0.133		0.279		
*p*-value	<0.001	0.005		<0.001		
**Convergent Analysis of Knowledge, Attitude, and Practice Scales**						
	**Knowledge scale**			**Attitude scale**		**Practice scale**
**Knowledge**				0.070		0.259
*p*-value				0.138		<0.001
**Attitude**	0.070					−0.275
*p*-value	0.138					<0.001

**Table 5 tropicalmed-08-00236-t005:** Bivariate analysis taking the KAP scales toward cholera as the dependent variable.

	Knowledge	Attitude	Practice
	Mean ± SD	Mean ± SD	Mean ± SD
**Sample status**			
General population	21.15 ± 5.72	72.33 ± 11.87	63.31 ± 10.18
Student healthcare professional	23.59 ± 4.46	71.89 ± 11.54	63.69 ± 10.80
Healthcare professional	24.45 ± 4.455	75.19 ± 12.77	64.20 ± 11.55
*p*-value	**<0.001 ***	**0.031 ***	0.756
**Gender**			
Female	23.93 ± 4.65	73.49 ± 12.27	64.85 ± 11.05
Male	21.48 ± 5.82	73.82 ± 12.44	60.96 ± 10.28
*p*-value	**<0.001 ***	0.805	**0.001 ***
**Marital status**			
Single/Widowed/divorced	22.77 ± 5.36	73.25 ± 12.17	63.37 ± 11.01
Married	24.38 ± 4.29	74.30 ± 12.60	64.79 ± 10.85
*p*-value	**0.002 ***	0.402	0.205
**Current governorate**			
Beirut	22.68 ± 5.48	73.12 ± 11.95	62.06 ± 10.30
Mont Lebanon	23.75 ± 4.90	73.43 ± 12.15	65.26 ± 10.90
North	22.95 ± 4.82	72.25 ± 13.75	62.34 ± 11.75
South	24.18 ± 5.00	75.44 ± 12.61	64.02 ± 12.22
Bekaa	21.66 ± 4.14	77.16 ± 12.12	66.77 ± 9.74
*p*-value	0.125	0.489	0.057
**Education level**			
School level	19.25 ± 8.12	70.41 ± 13.76	62.58 ± 12.85
University level	23.39 ± 4.95	73.67 ± 12.27	63.85 ± 10.93
*p*-value	0.106	0.367	0.692
**Education level of the head of the family**			
School level	21.89 ± 5.87	73.41 ± 11.89	64.18 ± 10.69
University level	23.85 ± 4.63	73.65 ± 12.49	63.67 ± 11.10
*p*-value	**0.001 ***	0.849	0.656
**Occupation**			
Unemployed	22.63 ± 4.90	72.99 ± 12.01	63.02 ± 10.67
Employed	23.88 ± 5.21	74.13 ± 12.57	64.56 ± 11.22
*p*-value	**0.009 ***	0.329	0.139
**Household monthly income**			
No income	22.58 ± 4.68	73.55 ± 12.68	63.90 ± 11.67
Low income: below 6 million	22.72 ± 5.12	70.90 ± 10.94	63.00 ± 10.22
Intermediate income: 6–18 million	22.91 ± 5.40	73.20 ± 12.23	63.53 ± 11.27
High income: over 18 million	24.43 ± 5.02	75.23 ± 12.54	64.39 ± 10.55
*p*-value	**0.011 ***	0.111	0.833
**Living place**			
Urban	23.24 ± 5.32	73.56 ± 11.91	63.39 ± 10.88
Rural	23.36 ± 4.47	73.64 ± 13.31	64.92 ± 11.17
*p*-value	0.823	0.955	0.186
**Housing condition**			
Very clean/Relatively clean	23.37 ± 5.11	73.34 ± 12.50	64.07 ± 11.12
Very dirty/Relatively dirty	22.54 ± 4.95	75.45 ± 10.51	61.84 ± 9.65
*p*-value	0.277	0.193	0.171
**Work status**			
Does not apply	22.54 ± 4.90	72.80 ± 12.05	63.04 ± 10.82
Full time	24.18 ± 4.72	74.03 ± 12.49	64.90 ± 10.63
Part-time	22.97 ± 5.99	74.34 ± 12.49	63.20 ± 11.95
*p*-value	**0.007 ***	0.518	0.231
**Cigarette Smoking status**			
Yes	21.33 ± 6.63	72.60 ± 13.34	61.66 ± 11.38
No	23.60 ± 4.73	73.74 ± 12.13	64.17 ± 10.88
*p*-value	**0.011 ***	0.495	0.093
**Having a chronic disease**			
Yes	23.54 ± 5.40	72.81 ± 11.97	65.96 ± 12.04
No	23.22 ± 5.03	73.75 ± 12.38	63.36 ± 10.69
*p*-value	0.614	0.538	0.056
**Easy access to healthcare**			
Yes	23.75 ± 4.94	73.97 ± 12.23	64.06 ± 10.91
No	20.90 ± 5.25	71.60 ± 12.55	62.59 ± 11.25
*p*-value	**<0.001 ***	0.131	0.292
**Cholera vaccines are available and effective**			
Yes	24.46 ± 4.24	74.41 ± 12.06	65.62 ± 11.05
No	21.93 ± 5.64	72.63 ± 12.53	61.76 ± 10.54
*p*-value	**<0.001 ***	0.127	**<0.001**
	**Correlation coefficient**	**Correlation coefficient**	**Correlation coefficient**
**Age**	0.081	0.074	0.073
*p*-value	0.086	0.117	0.122
**Household crowding index**	−0.101	−0.092	−0.023
*p*-value	**0.033 ***	0.053	0.633
**Financial well-being scale**	0.067	0.045	0.029
*p*-value	0.154	0.340	0.537

* Numbers in bold: significant results (*p*-values < 0.05).

**Table 6 tropicalmed-08-00236-t006:** Association between the source of information about cholera and the KAP scales.

	Knowledge	Attitude	Practice
	Mean ± SD	Mean ± SD	Mean ± SD
**Get Information about Cholera**			
**Social Media Platforms**			
Yes	23.36 ± 4.43	72.68 ± 12.22	63.37 ± 10.67
No	22.99 ± 6.96	76.70 ± 12.12	65.37 ± 11.88
*p*-value	0.611	0.004	0.110
**Electronic databases**			
Yes	23.88 ± 4.25	72.95 ± 12.26	64.19 ± 11.09
No	22.17 ± 6.23	74.74 ± 12.34	63.14 ± 10.76
*p*-value	0.002	0.143	0.335
**Physician**			
Yes	23.63 ± 4.61	73.05 ± 12.00	63.56 ± 11.32
No	22.68 ± 5.79	74.47 ± 12.78	64.25 ± 10.38
*p*-value	0.074	0.236	0.520
**Pharmacist**			
Yes	23.91 ± 4.22	73.89 ± 12.43	63.73 ± 10.96
No	21.75 ± 6.52	72.83 ± 12.00	64.04 ± 11.05
*p*-value	0.001	0.404	0.783
**Newspapers**			
Yes	22.98 ± 4.33	72.11 ± 12.54	62.09 ± 11.44
No	23.44 ± 5.48	74.42 ± 12.11	64.81 ± 10.59
*p*-value	0.327	0.055	0.012
**Schools and Universities**			
Yes	23.54 ± 4.34	73.02 ± 12.20	64.12 ± 10.80
No	22.64 ± 6.57	74.96 ± 12.48	63.07 ± 11.38
*p*-value	0.155	0.130	0.358
**Ministry of Public Health brochures**			
Yes	23.94 ± 4.48	72.53 ± 12.41	65.10 ± 10.74
No	21.96 ± 5.93	75.67 ± 11.86	61.28 ± 11.03
*p*-value	<0.001	0.011	<0.001
**Conferences**			
Yes	23.94 ± 4.37	73.16 ± 12.60	64.08 ± 11.39
No	22.23 ± 5.92	74.24 ± 11.82	63.41 ± 10.29
*p*-value	0.001	0.365	0.529
**I refuse to seek such information**			
Yes	21.58 ± 4.52	77.92 ± 13.84	57.38 ± 10.45
No	23.49 ± 5.13	73.04 ± 12.00	64.63 ± 10.78
*p*-value	0.012	0.020	<0.001

**Table 7 tropicalmed-08-00236-t007:** Multivariable linear regression analysis.

	UB	SB	*p*-Value	Confidence Interval	
				Lower Bound	Upper Bound
**Model 1: Taking the Knowledge Total Score as the Dependent Variable**					
**Cholera vaccines are available and effective (Yes vs. No *)**	1.649	0.165	<0.001	0.784	2.514
**Gender (Female vs. Male *)**	1.749	0.155	0.001	0.765	2.733
**Easy access to healthcare (Yes vs. No *)**	2.051	0.152	<0.001	0.926	3.177
**Work status (being a healthcare professional vs. other *)**	2.508	0.251	<0.001	1.514	3.501
**Work status (being a health student vs. other *)**	2.394	0.202	<0.001	1.183	3.605
**Work status (full-time vs. other *)**	1.243	0.122	0.009	0.317	2.170
**Get information about cholera (electronic databases vs. others *)**	1.191	0.114	0.008	0.314	2.068
**Get information about cholera (refuse to seek information vs. others *)**	−1.582	−0.100	0.020	−2.918	−0.246
**Cigarette smoking (Yes vs. No *)**	−1.314	−0.092	0.039	−2.562	−0.067
Variables entered in the model: age, household crowding index, marital status, gender, education level, easy access to healthcare, obtaining information from (electronic, physician, pharmacist, school/university, MOPH, conference, refuse to seek info), monthly income, work status, cholera vaccine is effective, cigarette smoking, governorate, sample status, and financial well-being scale.					
**Model 2: Linear regression taking the Attitude total score as the dependent variable.**					
**Get information about cholera (social media vs. others *)**	−3.290	−0.111	0.018	−6.019	−0.561
**Get information about cholera (MOPH vs. other *)**	−3.605	−0.138	0.003	−6.010	−1.199
**Work status (healthcare workers vs. others *)**	2.698	0.109	0.020	0.423	4.974
**Household crowding index**	−2.241	−0.095	0.043	−4.412	−0.071
Variables entered in the model: knowledge score, Household crowding index, age, sample status, easy access to healthcare, obtaining information from (social media, electronics, newspaper, school/university, MOPH), monthly income, cholera vaccine is effective and housing condition.					
**Model 3: Linear regression taking the Practice total score as the dependent variable.**					
**Attitude scale**	−0.278	−0.312	<0.001	−0.353	−0.202
**Knowledge scale**	0.432	0.201	<0.001	0.240	0.624
Get information (newspaper vs. other *)	−3.371	−0.148	0.001	−5.417	−1.325
Cholera vaccine is effective (Yes vs. No *)	3.177	0.145	0.001	1.271	5.083
Get information (MOPH vs. other *)	2.416	0.104	0.022	0.346	4.487
Gender (female vs. male *)	2.555	0.103	0.018	0.439	4.670
**Get information (social media vs. others)**	−2.479	−0.094	0.033	−4.759	−0.200
Variables entered in the model: knowledge, attitude, age, gender, living place, occupation, chronic disease, obtaining information (social media, newspaper, MOPH), housing condition, cholera vaccine is effective, cigarette smoking, and governorate.					

* Reference group.

## Data Availability

The datasets used and/or analyzed during the current study are available from the corresponding author upon reasonable request.

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
