# Peer review of "Knowledge, Attitude, and Practices of the General Population toward the Old-New Outbreak of Cholera in a Developing Country"

_tropicalmed, 2023, doi:10.3390/tropicalmed8040236_

Round 1

Reviewer 1 Report

The authors have assessed and analyzed the residents' knowledge, attitude, practices (KAP) based on a questionnaire survey for the purpose of preventing cholera in Lebanon. The authors have presented data that provide the basis for effective information distribution and awareness-raising measures for cholera prevention. This method is considered effective not only in Lebanon but also in cholera-endemic areas.

Although the age distribution is not shown in this manuscript, have the authors considered the increase in knowledge and awareness with age? In particular, have you considered the information literacy and education levels and KAPs of children and the elderly, who are more susceptible to infectious diseases?

Women make up 3/4 of the participants, but if, as the authors point out, women are more concerned about their own and their family's health, then the results of this survey may be quite biased. It cannot be generalized as all Lebanese people, can it?

Even if almost everyone in Lebanon has a smartphone, if you didn't include people who were not interested or cooperative in these kinds of surveys, was it possible that the correlation between interest and concern for infectious diseases and KAPs was very different from KAPs for people who were not interested or concerned?

It may be more important to increase access to digital resources and to expand correct information to push-type social media rather than the web, etc., in order to increase public awareness.

Could it be that even in areas with a higher house crowding index, if water and sewage infrastructure is improved, it would be more effective in controlling enteric infections than one might expect from KAP?

Author Response

Dear Editor,

Thank you for sharing the reviewer’s comments. We would like to thank the editorial board and the reviewers for the careful and thorough reading of this manuscript, and for their considerate comments and constructive suggestions.

We have carefully reviewed the comments and have revised the manuscript accordingly. Our responses are given in a point-by-point manner below. The modifications done are shown in track changes in the manuscript. The revision has been developed in consultation with all coauthors, and each coauthor has given approval to the final form of this revision.

We hope that the responses are satisfactory and that the revised manuscript is now suitable for publication. We look forward to hearing from you soon.

In what follows, the replies follow the reviewers’ comments. The responses are preceded by the term "authors' responses" and formatted in blue color (attached file).

Reviewer 1

Comments and Suggestions for Authors

The authors have assessed and analyzed the residents' knowledge, attitude, practices (KAP) based on a questionnaire survey for the purpose of preventing cholera in Lebanon. The authors have presented data that provide the basis for effective information distribution and awareness-raising measures for cholera prevention. This method is considered effective not only in Lebanon but also in cholera-endemic areas.

Authors’ Response: We thank the reviewer for acknowledging our work. We also highly appreciate the supportive ideas presented by the reviewer which will, indeed, increase our impact and reach.

Although the age distribution is not shown in this manuscript, have the authors considered the increase in knowledge and awareness with age? In particular, have you considered the information literacy and education levels and KAPs of children and the elderly, who are more susceptible to infectious diseases?

Authors’ Response: We thank the reviewer for this comment. We have added in the method section - study designs a sentence notifying that we have included all adults aged 18 years and more. Also, the age distribution was added in the first paragraph at the beginning of the results. We have already considered the association between age and KAPs in Table 5 bivariate analysis, the results showed that no association was found between age and KAPs (p-value>0.05). We do not have children in our sample; however, based on the reviewer request we have split the age into categories according to quartile and we have run an analysis between education level and KAPs according to age categories. A paragraph was added in the results section bivariate analysis as follows:

“No association was found between age and KAP (p>0.05 for all). The age variable was divided into quartiles (Category 1 below 22 years, Category 2 between 23 – 25 years, Category 3 between 26 – 35 years, and Category 4 above 36 years), and bivariate analysis was conducted between age categories and KAP. The results showed a significant association between education level and attitudes and practices in participants below 22 years. A significantly higher mean attitude score was found in those with a school education level as compared to a university education level (80.50 vs. 72.09, p<0.001), while a higher mean practice score was found in those with a university education level as compared to school education level (63.27 vs. 54.50, p<0.001). The association between knowledge scale and education level tended to be significant in participants over 36 years (p=0.055). No significant association was found between education level and KAP in other age categories.”

Women make up 3/4 of the participants, but if, as the authors point out, women are more concerned about their own and their family's health, then the results of this survey may be quite biased. It cannot be generalized as all Lebanese people, can it?

Authors’ Response: We thank the reviewer for this comment. Yes, the reviewer is right we cannot generalize to all Lebanese people therefore this idea was added to the limitation section as follows:

“Also, the findings of the current study could not represent the general Lebanese population as the sample enrolled unequal gender distribution where the majority were females”

Even if almost everyone in Lebanon has a smartphone, if you didn't include people who were not interested or cooperative in these kinds of surveys, was it possible that the correlation between interest and concern for infectious diseases and KAPs was very different from KAPs for people who were not interested or concerned?

Authors’ Response: We thank the reviewer for this comment. It is a good point; however, we did not include in our study those kind of people. The idea was added in the limitation section as follows:

“However, some other people who were not interested or cooperative in these kinds of surveys were not enrolled in the study; this might have affected the results as a possible correlation might be find between those interested and concerned for infectious diseases and KAPs and those who were not”.

It may be more important to increase access to digital resources and to expand correct information to push-type social media rather than the web, etc., in order to increase public awareness.

Authors’ Response: We thank the reviewer for this comment. This idea was added to the “Practical implications” paragraph as follows:

“In order to increase public awareness, it may be more important to increase access to digital resources and to expand correct information to push-type social media rather than the web.”

Could it be that even in areas with a higher house crowding index, if water and sewage infrastructure is improved, it would be more effective in controlling enteric infections than one might expect from KAP?

Authors’ Response: We thank the reviewer for this comment. This idea was added to the “Practical implications” paragraph as follows:

“Also, in areas with a higher house crowding index, if water and sewage infrastructure is improved, it would be more effective in controlling enteric infections than enhancing KAP.”

We hope, by the above, to satisfy the reviewer’s concerns and to have this manuscript eligible for publication in your Journal. While we remain available for further inquiries or suggestions from your side, we look forward to hearing from you at your earliest convenience.

Yours sincerely,

The corresponding author

Dr. Jihan Safwan, on behalf of all authors

Reviewer 2 Report

Dear authors,

Please read my suggestions for improvement.

ABSTRACT

The abstract provides a clear overview of the study's background, objectives, methods, results, and conclusions. It outlines the context of the cholera outbreak in Lebanon and the need to assess the general population's knowledge, attitudes, and practices. The use of an online cross-sectional study with snowball sampling is mentioned, along with the number of participants.

The results section in the abstract highlights the key findings, including the validity and reliability of the KAP scales and various factors associated with different aspects of the KAP assessment. The conclusion emphasizes the need for public health stakeholders and governmental authorities to take action based on the identified gaps in knowledge, attitudes, and practices.

However, there are a few suggestions for improvement:

- Consider providing more context in the background about the significance of the cholera outbreak in Lebanon and why assessing KAP is essential.

- In the methods section, briefly mention the development and validation process of the KAP tool.

- Ensure the beta values are correctly formatted with appropriate spacing in the results section.

- The conclusion could mention specific actions or recommendations that can be derived from the study findings to improve public health efforts in curbing disease transmission.

INTRODUCTION

The introduction provides a comprehensive background on cholera, its global impact, and the recent outbreak in Lebanon. It highlights the challenges faced by the Lebanese healthcare system, the country's economic crisis, and the need for immediate action to control the cholera outbreak. The importance of understanding the general population's knowledge, attitudes, and practices (KAP) is also discussed. Lastly, the introduction clearly states the study's aim and relevance in guiding prevention and awareness strategies.

However, there are a few suggestions for improvement:

·        The mention of "cholera virus" in line 47 should be corrected to "Vibrio cholerae bacteria" since bacteria, not a virus, cause cholera.

·        In line 85, consider rephrasing "A recently published study" to "A study conducted in Lebanon" or provide more context about the specific study to emphasize its relevance.

·        In the final paragraph, it would be helpful to elaborate on the limitations of the previous study and the specific contributions this study aims to make in addressing those limitations.

·        Consider discussing the relevance of KAP assessments in managing infectious disease outbreaks and how it has been used in other contexts. This could provide further justification for the study.

MATERIAL AND METHODS

The Materials and Methods section provides a clear and detailed description of the study design, ethical considerations, sample size calculation, questionnaire development, translation procedure, and statistical analysis. Using a cross-sectional study design and snowball sampling technique allows for a diverse sample from different regions of Lebanon. The inclusion of validated scales and the use of Cronbach's alpha to assess internal consistency strengthens the credibility of the study.

However, there are a few points to consider for improvement:

·        In line 97, the wording "which helped reach people in remote areas where cholera was first detected" could be rephrased to emphasize the advantages of using an online questionnaire in reaching a broader population.

·        The snowball sampling technique, while useful in specific contexts, may introduce selection bias. It would be helpful to acknowledge this limitation in the methodology section.

·        In lines 127-130 and 134-138, it is mentioned that the InCharge Financial Distress-Financial Well-Being (IFDFW) Scale and the Knowledge, Attitude, and Practice scales were used in the study. It would be helpful to briefly discuss the rationale for choosing these scales in the context of the study.

·        In line 160, consider mentioning the sample size of the bilingual participants in the pilot study for the final questionnaire.

·        While using principal component analysis (PCA) for factor analysis is appropriate, consider mentioning the rotation method (e.g., varimax or promax).

RESULTS/DISCUSSION

The results seem fine. The discussion in the study provides valuable insights into the knowledge, attitude, and practice (KAP) towards cholera in the Lebanese community. However, some areas could be improved or expanded upon:

·        More emphasis on limitations: While the study acknowledges some limitations, it could provide a more in-depth discussion of the potential impact of these limitations on the study's results and conclusions. For instance, the cross-sectional design's inherent limitations regarding temporality and causality could be discussed.

·        Discussing the role of socio-economic factors: The study could delve deeper into how socio-economic factors shape KAP towards cholera. It would be valuable to explore how factors such as income, education level, and access to resources affect the KAP of individuals.

·        Addressing misinformation and disinformation: The study touches upon the fact that worse practices were significantly associated with getting information from social media. However, it could benefit from a more comprehensive discussion on how to address misinformation and disinformation in the context of health education.

·        Expanding on the practical implications: The discussion provides some practical implications, but it could benefit from a more detailed description of specific strategies and interventions that public health stakeholders could implement to improve KAP towards cholera in the Lebanese community.

·        Comparisons to other populations: The study compares the Lebanese community's KAP to other countries dealing with cholera outbreaks. However, a more detailed comparison could provide additional insights into the similarities and differences between populations and how that might inform future interventions.

·        Inclusion of mental health factors: The study does not explore the impact of mental health on KAP towards cholera. Including this aspect in future research could provide a more holistic understanding of the factors influencing individuals' KAP and help inform targeted interventions for different sub-groups within the community.

CONCLUSION

The conclusion of the study provides a concise summary of the main findings. However, some aspects could be improved or expanded upon:

·        Elaborating on the implications: The conclusion could benefit from a more detailed discussion of the practical implications of the study's findings, such as specific actions or recommendations for public health stakeholders and governmental authorities to improve cholera KAP in the Lebanese community.

·        Expanding on the significance of the findings: The conclusion could provide more context on the significance of the study's findings within the broader literature on cholera KAP and emphasize the novelty of the study's tools and methodology.

·        Highlighting limitations and future research directions: The conclusion could briefly mention the study's limitations and suggest avenues for future research to address these limitations, such as exploring the impact of mental health on cholera KAP and conducting longitudinal studies to establish causality.

·        Discussing potential interventions: The conclusion could propose interventions that effectively address the gaps identified in the study, such as targeted health education campaigns, public awareness programs, and strategies to combat misinformation and disinformation on social media.

·        Emphasizing the importance of a multi-faceted approach: The conclusion could stress the need for a multi-faceted approach to improving cholera KAP, incorporating strategies targeting knowledge, attitudes, and practices, as well as considering socio-economic factors and information sources.

By addressing these points, the conclusion can provide a more comprehensive summary of the study's findings and implications, offering valuable insights for public health stakeholders and governmental authorities working to improve cholera KAP and curb disease transmission.

Author Response

Dear Editor,

Thank you for sharing the reviewer’s comments. We would like to thank the editorial board and the reviewers for the careful and thorough reading of this manuscript, and for their considerate comments and constructive suggestions.

We have carefully reviewed the comments and have revised the manuscript accordingly. Our responses are given in a point-by-point manner below. The modifications done are shown in track changes in the manuscript. The revision has been developed in consultation with all coauthors, and each coauthor has given approval to the final form of this revision.

We hope that the responses are satisfactory and that the revised manuscript is now suitable for publication. We look forward to hearing from you soon.

In what follows, the replies follow the reviewers’ comments. The responses are preceded by the term "authors' responses" and formatted in blue color (attached file).

Reviewer 2

Comments and Suggestions for Authors

Dear authors,

Please read my suggestions for improvement.

ABSTRACT

The abstract provides a clear overview of the study's background, objectives, methods, results, and conclusions. It outlines the context of the cholera outbreak in Lebanon and the need to assess the general population's knowledge, attitudes, and practices. The use of an online cross-sectional study with snowball sampling is mentioned, along with the number of participants.

The results section in the abstract highlights the key findings, including the validity and reliability of the KAP scales and various factors associated with different aspects of the KAP assessment. The conclusion emphasizes the need for public health stakeholders and governmental authorities to take action based on the identified gaps in knowledge, attitudes, and practices.

Authors’ Response: We thank the reviewer for acknowledging our work. We also highly appreciate the supportive ideas presented by the reviewer which will, indeed, increase our impact and reach.

However, there are a few suggestions for improvement:

  • Consider providing more context in the background about the significance of the cholera outbreak in Lebanon and why assessing KAP is essential.

Authors’ Response: We thank the reviewer for this comment. A sentence was added to the background within the abstract that provides more context about the significance of the cholera outbreak in Lebanon and why assessing KAP is essential.

  • In the methods section, briefly mention the development and validation process of the KAP tool.

Authors’ Response: We thank the reviewer for this comment. More details were added to the methods section.

  • Ensure the beta values are correctly formatted with appropriate spacing in the results section.

Authors’ Response: We thank the reviewer for this comment. The word beta has been capitalized as “Beta” in the abstract. This way it will ensure the standardization of reporting within the whole document. Correct formatting and appropriate spacing in the results have been ensured.

  • The conclusion could mention specific actions or recommendations that can be derived from the study findings to improve public health efforts in curbing disease transmission.

Authors’ Response: We thank the reviewer for this comment. We added a sentence to the conclusion that addresses specific actions or recommendations to improve public health efforts in curbing disease transmission.

INTRODUCTION

The introduction provides a comprehensive background on cholera, its global impact, and the recent outbreak in Lebanon. It highlights the challenges faced by the Lebanese healthcare system, the country's economic crisis, and the need for immediate action to control the cholera outbreak. The importance of understanding the general population's knowledge, attitudes, and practices (KAP) is also discussed. Lastly, the introduction clearly states the study's aim and relevance in guiding prevention and awareness strategies.

Authors’ Response: We thank the reviewer for acknowledging our work. We also highly appreciate the supportive ideas presented by the reviewer which will, indeed, increase our impact and reach.

However, there are a few suggestions for improvement:

  • The mention of "cholera virus" in line 47 should be corrected to "Vibrio choleraebacteria" since bacteria, not a virus, cause cholera.

Authors’ Response: We thank the reviewer for this important comment. The word virus has been changed to bacteria and the name of the bacteria "Vibrio cholerae bacteria” is already mentioned in that sentence.

  • In line 85, consider rephrasing "A recently published study" to "A study conducted in Lebanon" or provide more context about the specific study to emphasize its relevance.

Authors’ Response: We thank the reviewer for this comment. We rephrased the sentence as recommended by the reviewer.

  • In the final paragraph, it would be helpful to elaborate on the limitations of the previous study and the specific contributions this study aims to make in addressing those limitations.

Authors’ Response: We thank the reviewer for this comment. This point has already been addressed in line 90 of the introduction; we elaborated it.

  • Consider discussing the relevance of KAP assessments in managing infectious disease outbreaks and how it has been used in other contexts. This could provide further justification for the study.

Authors’ Response: We thank the reviewer for this comment. Concept added to the introduction.

MATERIAL AND METHODS

The Materials and Methods section provides a clear and detailed description of the study design, ethical considerations, sample size calculation, questionnaire development, translation procedure, and statistical analysis. Using a cross-sectional study design and snowball sampling technique allows for a diverse sample from different regions of Lebanon. The inclusion of validated scales and the use of Cronbach's alpha to assess internal consistency strengthens the credibility of the study.

Authors’ Response: We thank the reviewer for acknowledging our work. We also highly appreciate the supportive ideas presented by the reviewer which will, indeed, increase our impact and reach.

However, there are a few points to consider for improvement:

  • In line 97, the wording "which helped reach people in remote areas where cholera was first detected" could be rephrased to emphasize the advantages of using an online questionnaire in reaching a broader population.

Authors’ Response: We thank the reviewer for this comment. The sentence was rephrased to emphasize the advantages of using an online questionnaire in reaching a broader population.

  • The snowball sampling technique, while useful in specific contexts, may introduce selection bias. It would be helpful to acknowledge this limitation in the methodology section.

Authors’ Response: We thank the reviewer for this comment. The idea was added to the limitation section as follows:

“Furthermore, data were collected using a snowball technique (an online questionnaire), which is a is a nonrandom technique that could have been associated with a possible selection bias as this data collection technique might have eliminated people with poor digital literacy”.

  • In lines 127-130 and 134-138, it is mentioned that the InCharge Financial Distress-Financial Well-Being (IFDFW) Scale and the Knowledge, Attitude, and Practice scales were used in the study. It would be helpful to briefly discuss the rationale for choosing these scales in the context of the study.

Authors’ Response: We thank the reviewer for this comment. As suggested by the reviewer, a brief discussion of the rationale for choosing the IFDFW and KAP scales in the context of the study.

  • In line 160, consider mentioning the sample size of the bilingual participants in the pilot study for the final questionnaire.

Authors’ Response: We thank the reviewer for this comment. The sample size was added.

  • While using principal component analysis (PCA) for factor analysis is appropriate, consider mentioning the rotation method (e.g., varimax or promax).

 Authors’ Response: We thank the reviewer for this comment. A sentence was added to the “Statistical analysis” part mentioning the type of rotation as follows: “The Promax rotation method was used”.

RESULTS/DISCUSSION

The results seem fine. The discussion in the study provides valuable insights into the knowledge, attitude, and practice (KAP) towards cholera in the Lebanese community. However, some areas could be improved or expanded upon:

Authors’ Response: We thank the reviewer for acknowledging our work. We also highly appreciate the supportive ideas presented by the reviewer which will, indeed, increase our impact and reach.

  • More emphasis on limitations: While the study acknowledges some limitations, it could provide a more in-depth discussion of the potential impact of these limitations on the study's results and conclusions. For instance, the cross-sectional design's inherent limitations regarding temporality and causality could be discussed.

Authors’ Response: We thank the reviewer for this comment. A sentence was already written in the limitation section discussing this point as follows:

“The cross-sectional design does not provide temporality but remains practical for capturing a snapshot amid outbreaks and rapidly identifying gaps, although causal associations cannot be confirmed”. Other explanations were also provided.

  • Discussing the role of socio-economic factors: The study could delve deeper into how socio-economic factors shape KAP towards cholera. It would be valuable to explore how factors such as income, education level, and access to resources affect the KAP of individuals.

Authors’ Response: We thank the reviewer for this comment. The association between socio economic factors was already studied in Table 5 in the bivariate analysis.

  • Addressing misinformation and disinformation: The study touches upon the fact that worse practices were significantly associated with getting information from social media. However, it could benefit from a more comprehensive discussion on how to address misinformation and disinformation in the context of health education.

Authors’ Response: This idea was added in the discussion.

  • Expanding on the practical implications: The discussion provides some practical implications, but it could benefit from a more detailed description of specific strategies and interventions that public health stakeholders could implement to improve KAP towards cholera in the Lebanese community.

Authors’ Response: We thank the reviewer for this comment. The practical implications were elaborated.

  • Comparisons to other populations: The study compares the Lebanese community's KAP to other countries dealing with cholera outbreaks. However, a more detailed comparison could provide additional insights into the similarities and differences between populations and how that might inform future interventions.

Authors’ Response: We thank the reviewer for this comment. This point was elaborated in the discussion.

  • Inclusion of mental health factors: The study does not explore the impact of mental health on KAP towards cholera. Including this aspect in future research could provide a more holistic understanding of the factors influencing individuals' KAP and help inform targeted interventions for different sub-groups within the community.

Authors’ Response: We thank the reviewer for this comment. This idea was added to the limitation section as follows:

“In addition, the study did not explore the impact of mental health on KAP towards cholera as this aspect could provide a more holistic understanding of the factors influencing individuals' KAP and help inform targeted interventions for different sub-groups within the community”

CONCLUSION

The conclusion of the study provides a concise summary of the main findings. However, some aspects could be improved or expanded upon:

Authors’ Response: We thank the reviewer for acknowledging our work. We also highly appreciate the supportive ideas presented by the reviewer which will, indeed, increase our impact and reach.

  • Elaborating on the implications: The conclusion could benefit from a more detailed discussion of the practical implications of the study's findings, such as specific actions or recommendations for public health stakeholders and governmental authorities to improve cholera KAP in the Lebanese community.

Authors’ Response: We thank the reviewer for this comment. Public health actions were further elaborated.

  • Expanding on the significance of the findings: The conclusion could provide more context on the significance of the study's findings within the broader literature on cholera KAP and emphasize the novelty of the study's tools and methodology.

Authors’ Response: We thank the reviewer for this comment. We elaborated on the significance of study findings in the current context of socioeconomic hardship and sanitary decay, which might increase the usefulness of the suggested tools in developing countries with similar circumstances.

  • Highlighting limitations and future research directions: The conclusion could briefly mention the study's limitations and suggest avenues for future research to address these limitations, such as exploring the impact of mental health on cholera KAP and conducting longitudinal studies to establish causality.

Authors’ Response: We thank the reviewer for this comment. Additional limitations and suggestions related to the mental health were also added.

  • Discussing potential interventions: The conclusion could propose interventions that effectively address the gaps identified in the study, such as targeted health education campaigns, public awareness programs, and strategies to combat misinformation and disinformation on social media.

Authors’ Response: We thank the reviewer for this comment. Further suggestions were added and elaborated in the discussion.

  • Emphasizing the importance of a multi-faceted approach: The conclusion could stress the need for a multi-faceted approach to improving cholera KAP, incorporating strategies targeting knowledge, attitudes, and practices, as well as considering socio-economic factors and information sources.

Authors’ Response: We thank the reviewer for this comment. Ideas added to the conclusion.

  • By addressing these points, the conclusion can provide a more comprehensive summary of the study's findings and implications, offering valuable insights for public health stakeholders and governmental authorities working to improve cholera KAP and curb disease transmission.

Authors’ Response: We thank the reviewer for this comment. Thank you, indeed!

We hope, by the above, to satisfy the reviewer’s concerns and to have this manuscript eligible for publication in your Journal. While we remain available for further inquiries or suggestions from your side, we look forward to hearing from you at your earliest convenience.

Yours sincerely,

The corresponding author

Dr. Jihan Safwan, on behalf of all authors
